# Clinical Care Conditions and Needs of Palliative Care Patients from Five Italian Regions: Preliminary Data of the DEMETRA Project

**DOI:** 10.3390/healthcare8030221

**Published:** 2020-07-20

**Authors:** Gianlorenzo Scaccabarozzi, Emanuele Amodio, Luca Riva, Oscar Corli, Marco Maltoni, Grazia Di Silvestre, Adriana Turriziani, Piero Morino, Giacomo Pellegrini, Matteo Crippa

**Affiliations:** 1Dipartimento Fragilità/Rete Locale Cure palliative, ASST Lecco, 23807 Merate (LC), Italy; g.scaccabarozzi@asst-lecco.it; 2Department of Health Promotion, Mother and Child Care, Internal Medicine and Medical Specialties (PROMISE), University of Palermo, 90133 Palermo, Italy; emanuele.amodio@unipa.it; 3UOS Unità Cure Palliative Ospedaliere, ASST Lecco, 23900 Lecco, Italy; lu.riva@asst-lecco.it; 4Pain and Palliative Care Research Unit, Istituto di Ricerche Farmacologiche Mario Negri-IRCCS, 20156 Milan, Italy; oscar.corli@marionegri.it; 5Palliative Care Unit, Istituto Scientifico Romagnolo per lo Studio e la Cura dei Tumori (IRST) IRCCS, 47014 Meldola (FC), Italy; marcocesare.maltoni@auslromagna.it; 6UOS Cure Palliative ASP Palermo, 90129 Palermo, Italy; g.disilvestre.gds@gmail.com; 7Master Cure Palliative, Università Cattolica S.Cuore, 00168 Rome, Italy; adriana.turriziani@gmail.com; 8UFC Coordinamento Aziendale Cure Palliative USL Toscana Centro, 50142 Firenze, Italy; piero.morino@uslcentro.toscana.it; 9Fondazione Floriani, Via privata Nino Bonnet, 2-20154 Milan, Italy; m.crippa@fondazionefloriani.eu

**Keywords:** adult, frail older, assessment of healthcare needs, network, palliative care, stress, emotional

## Abstract

In order to plan the right palliative care for patients and their families, it is essential to have detailed information about patients’ needs. To gain insight into these needs, we analyzed five Italian local palliative care networks and assessed the clinical care conditions of patients facing the complexities of advanced and chronic disease. A longitudinal, observational, noninterventional study was carried out in five Italian regions from May 2017 to November 2018. Patients who accessed the palliative care networks were monitored for 12 months. Sociodemographic, clinical, and symptom information was collected with several tools, including the Necesidades Paliativas CCOMS-ICO (NECPAL) tool, the Edmonton Symptom Assessment System (ESAS), and interRAI Palliative Care (interRAI-PC). There were 1013 patients in the study. The majority (51.7%) were recruited at home palliative care units. Cancer was the most frequent diagnosis (85.4%), and most patients had at least one comorbidity (58.8%). Cancer patients reported emotional stress with severe symptoms (38.7% vs. 24.3% in noncancer patients; *p* = 0.001) and were less likely to have clinical frailty (13.3% vs. 43.9%; *p* < 0.001). Our study confirms that many patients face the last few months of life with comorbidities or extreme frailty. This study contributes to increasing the general knowledge on palliative care needs in a high-income country.

## 1. Introduction

It is well known that in high-income countries there has been a continuous increase in the life expectancy of the general population, and for many people, the last period of life is characterized by complex chronic conditions or frailty [1]. It has also been estimated that each year between 69% and 82% of people who die could benefit from palliative care [2].

According to the definition of the World Health Organization (WHO), palliative care means “an approach that improves the quality of life of patients and their families facing the problems associated with life-threatening illness, through the prevention and relief of suffering by means of early identification and impeccable assessment and treatment of pain and other problems, physical, psychosocial and spiritual” [3]. About three-quarters of those with palliative care needs are patients with cancer, cardiovascular problems, or both. Currently, around 40 million people globally need palliative care, but only 14% of them actually have access to it [4].

In Italy, a growing interest in palliative care has been observed since the late 1990s. In 2001, palliative care assistance was introduced into the National Health Plan and subsequently into the Essential Levels of Assistance [5]. A further impetus was given by Italian Law 38/2010; in agreement with WHO recommendations, this law sanctioned the right of all citizens to access palliative care and established local palliative care networks to guarantee the continuity of care for the patient through the integration of hospital or hospice care and home care [6]. However, at the national level, there are currently significant differences in the density of palliative care networks as well as the number of provided services and even the types of patients and their families who benefit from these services [7].

The need to improve these networks is evident if we consider that approximately one-third of patients dying of cancer in Italy die in a hospital ward for acute patients, following an average hospital stay of about 12 days [8]. One of the needed improvements is the definition of programs that are as customized as possible, particularly in the last months of the disease, when there is a progressive loss of independence and an exacerbation of physical and mental symptoms, affecting not only the patient but also the family nucleus that faces this dramatic moment together [9].

In a theoretical model, palliative care networks take care of the needs of patients and their families in different settings: home, hospice, and hospital. However, there are no up-to-date, reliable, and complete data to describe the actual current activity. The mandatory data required by the Italian Ministry of Health are in fact often incomplete and, in any case, poorly detailed [10].

Based on these premises, we aimed to analyze the situation of palliative care in five Italian regions. This is the first work on the data collected by the DEMETRA 1 study. This work summarizes the primary objective of the research, which is to assess the clinical care conditions of patients with complex and advanced chronic conditions who access the local palliative care networks.

## 2. Materials and Methods

### 2.1. Study Design

The DEMETRA 1 study was conceived as a longitudinal, observational, noninterventional study aimed at assessing the clinical care conditions of patients facing the complexities of advanced and chronic disease who access the local palliative care networks. Five local palliative care networks (Lecco, Forlì, Florence, Rome, and Palermo) located in five Italian regions (Lombardy, Emilia-Romagna, Tuscany, Lazio, and Sicily, respectively) were involved.

Patients were recruited from three palliative care settings: home care (Lecco, Forlì, Florence, and Palermo); inpatient service (Lecco, Florence, and Rome); and hospice (Lecco, Forlì, Florence, and Palermo). The inclusion and exclusion criteria were defined in advance. To be eligible, patients had to have the following characteristics:New patients accessing the palliative care network during the recruitment period;Age 18 or over;Presence of a chronic disease with a progressive course of any nature requiring palliative intervention;Written consent for personal data processing and informed consent for participation in the study.

The exclusion criteria were as follows:Impossibility of ensuring regular follow-up (e.g., due to consent withdrawal);Already being in the care of a palliative care network at the start of recruitment;Transfer to a facility not included in the monitoring or external to the palliative care network (e.g., nursing home).

Patients who met the inclusion criteria were entered into the study until the sample size established for each research network was reached. They were monitored for 12 months.

### 2.2. Study Duration and Time Schedule

The study had an overall duration of 18 months, from May 2017 to November 2018. From May 2017 to November 2017, active patient recruitment was carried out, and patients were monitored until November 2018. Then, all data were processed and submitted to quality control for evaluating the completeness and consistency of the recorded data.

### 2.3. Data Collection

Data from forms completed for each patient were inserted into an electronic data collection tool (e-CRF). All data were anonymized and could not be tracked to the individuals participating in the study. Only one researcher of each unit had the key linking the anonymous code and the patient’s personal details.

The data were collected by health professionals (doctors, nurses, or social workers, depending on the type of information) at the patient’s intake by the palliative care network, which coincided with the patient’s entry into the study (day 0). Clinical data were also collected after one week (day 7) and, if the patient was still alive, every three months (day 90). If the patient died during the monitoring period, the date and place of death were also recorded.

For the collection of data, depending on the type of information required, some standardized international tools were used. The interRAI Palliative Care (interRAI-PC) assessment instrument, initially created in Canada and then spread all over the world, provides a standardized, comprehensive means to identify person-specific needs and supports clinicians in addressing important factors such as aspects of function, health, and social support [11]. The Personal Health Profile (PHP) at the study entry was assessed and subsequently reported on a multidimensional evaluation scale (interRAI), which provides values between 0 and 4. The analyzed variables were the presence or absence of certain parameters (vomiting, dehydration, loss of body weight, dyspnea, edema, decline in cognitive function) and the activities of daily life (ADLs).

The Necesidades Paliativas CCOMS-ICO (NECPAL) tool is a Spanish tool, developed in 2013 by the working group of the Catalan Institute of Oncology [12]. It is a quali-quantitative, multifactorial assessment tool. In NECPAL the question regarding surprise (“Would I be surprised if this patient died in the next 12 months?”) constitutes a discriminating element for the identification of patients with palliative care needs, while the 13 other items simply evaluate the presence or absence of clinical problems [13]. The specific “fragility” item of the NECPAL has been used to determine whether a patient is in a condition of extreme fragility. NECPAL’s main objective is the early identification of persons with palliative care needs and life-limiting prognosis in health and social services to actively improve the quality of their care.

Specific scales were filled in at 0, 7, and 90 days to collect information on the patient’s clinical status and psychological condition. The Edmonton Symptom Assessment System (ESAS), developed by E. Bruera almost 30 years ago [14], has been psychometrically validated and translated into over 20 languages. ESAS is composed of 11-point numeric rating scales (NRS) ranging from 0 (no symptom) to 10 (worst possible). It has been used to assess common symptoms including anxiety, depression, dyspnea, pain, malaise, nausea, sleepiness, and asthenia.

The psychosocial conditions of the patients and their caregivers were investigated through a brief questionnaire with seven items requiring a single answer (positive or negative) regarding the presence or absence of various conditions of psychosocial difficulty related to illness and assistance. Additional data collected are shown in Table 1.

### 2.4. Consent Procedure and Ethical Approval

When a patient entered the care of the palliative care network, a medical doctor or assistant informed them and their caregivers about the study, asking for their availability to join the study and their authorization to collect personal data. Written informed consent was obtained for all recruited patients. For patients without the capacity to consent, the designated consultee provided assent by signing a declaration form.

The study received the approval of the Ethics Committee of ASST Lecco (Local Social Health Authority) on 1 December 2016. Subsequently, all other ethics committees of the centers that joined the study provided approval of the study.

### 2.5. Sample Size and Data Analysis

The overall number of new cases enrolled per year in all study settings was taken as our reference population (3707 new cases per year). Considering a maximum error of 4% (95% confidence) as tolerable, we first calculated a sample size for each operating unit and then an overall sample size (*n* = 837) that could be considered representative of the entire reference population. This sample size was conservatively increased by 10% according to the potential loss to follow-up, and thus a minimum sample size of 921 patients was required.

Qualitative variables were summarized with absolute and relative (percentage) frequencies; for quantitative variables, means (± standard deviations) or medians (with interquartile ranges) were calculated depending on their parametric distribution. Statistical comparison for qualitative and quantitative variables was performed with the χ^2^ test and Student’s *t*-test or ANOVA, respectively. A *p*-value less than 0.05 was considered statistically significant. All statistical analyses were done with the statistical software R, version 3.6.1 (R Foundation for Statistical Computing, Vienna, Austria) [15].

## 3. Results

Of the 1013 patients included in the study and monitored, 531 were women and 482 were men, with a mean age of 75 years (Table 2). The majority of patients were recruited at home palliative care units. Most patients had a cancer diagnosis and at least one comorbidity. The most frequent cancer types were lung, pancreatic, and colorectal cancer. Among the noncancer diagnoses, cardiovascular diseases were the most frequent, followed by dementia and diseases of the respiratory system.

Table 3 relates the main demanding problems observed in patients who accessed the local palliative care networks to their diagnosis (cancer vs. other). ESAS was completed in 844 cases (83.3% of the total sample). This was due to the inability of some patients to report on their symptoms given their serious condition. Cancer patients were significantly younger, reported a period of emotional stress with persistent severe psychological symptoms, were less likely to have clinical frailty, and suffered more frequently from chronic pain.

Table 4 summarizes the same items in relation to the healthcare setting. Patients in hospice care were significantly more likely to have severe emotional stress, extreme or severe frailty, and depression. Patients receiving home palliative care more frequently had cancer and were suffering from pain, malaise, and nausea.

## 4. Discussion

This study summarizes the characteristics of patients with palliative care needs in a sample population that could be considered representative of the Italian general population. The primary aim of the study was to define the clinical profile of these patients from the moment they access the palliative care network, through the analysis of clinical aspects, both objective and subjective. This aim was conceived to effectively anticipate patients’ needs and plan palliative care services based on an understanding of how prevalent palliative-care-related problems are across diagnostic groups and healthcare settings.

Our estimates can be of paramount importance for the provision of palliative care in a rapidly changing healthcare context as seen in Italy and other high-income countries. It is well known that in such countries there is a continuous increase in the life expectancy of the population [16], and the last period of life can be characterized by complex chronic conditions or frailty [17].

The results of our study confirm these findings and suggest that about 60% of patients at the end of life are affected by comorbidities and severe emotional stress, whereas about 18% experience extreme or severe clinical frailty. Palliative care, as defined by the WHO, can offer patients a comprehensive response to suffering through an approach focused on the control of pain and other symptoms, with specific attention to quality of life. This is particularly important if we take into account that in our experience about two out of three patients reported pain at the first visit, and this was significantly more frequent in patients who were recruited in a home palliative care setting.

However, we found that the general clinical presentation was usually highly complex due to the presence of other significant symptoms such as malaise or asthenia, which affected about 80% and 90% of our patients, respectively, regardless of the healthcare setting in which they were recruited. Moreover, as reported by other authors, these symptoms start to deteriorate earliest and are most frequent in cancer patients [18], who made up 85% of the population in our study. A high percentage of patients experience multiple symptoms at the same time; these may result from a number of factors such as disease progression or metastases, drug treatment, or treatments for concurrent conditions.

Current trends [19] suggest that older people with progressive long-term conditions are increasingly prevalent amongst those in need of palliative care and, due to the interplay of multimorbid, long-term conditions and frailty, older people could have very different patterns of illness.

While for decades palliative care seemed to be the prerogative of cancer patients, it is now generally recognized that patients with all chronic progressive diseases and conditions can benefit from it. To date, however, noncancer patients have been underrepresented in palliative care centers [20]. It should be noted, on the other hand, that the prevalence of cancer in Italy is steadily increasing [21]; thus, an increase in cancer patients managed by the palliative care networks is to be expected.

Patients with cancer seem to be more vulnerable mentally, and, as observed in other studies [18,22,23,24], mental health issues such as anxiety and depressive symptoms are very frequent. As reported in dedicated studies [25], this makes it particularly important for palliative care paths to collect information on the psychological state of patients and caregivers so that adequate access to psychological and psychiatric services can be provided. The ESAS has been demonstrated to effectively assess the distressing symptoms impairing patients’ quality of life and can be utilized to evaluate the symptom status at baseline as well as the impact of symptom management strategies on the overall disease burden [26].

Finally, our study confirms that the different settings where the palliative care networks operate (home, hospital, and hospice) take care of patients with different, very distinct problems and palliative care needs. Moreover, the management of the most appropriate palliative care network can contribute to lowering hospitalization costs for end-of-life patients and reducing their probability of dying in hospitals [27].

This preliminary analysis of the DEMETRA 1 data may have some limitations. In particular, some missing responses in the ESAS questionnaires and a number of patients who were lost to follow-up may have introduced a potential selection bias. Moreover, despite the large sample size, we cannot be sure that the five Italian centers are fully representative of the Italian palliative care scenario, and the use of a convenience sample could affect representativeness with respect to the Italian population with palliative care needs. Last but not least, we do not have information about pharmacotherapy that, according to Abernethy et al. [28], could increase knowledge about clinical pathways of patients.

## 5. Conclusions

In spite of the possible limitations, this analysis contributes to the general knowledge on palliative care networks in a high-income country such as Italy where data to describe the actual activity carried out are often poor and not up to date. The complexity of palliative care patients strongly encourages us to increase the attention paid to the assessment of their preferences and desires. It should be noted that their closest contacts, whose burden in human and monetary terms (direct and indirect costs) remains to be clarified, also require attention, particularly in the assessment of their degree of satisfaction with the path taken by the assisted patient.

## Figures and Tables

**Table 1 healthcare-08-00221-t001:** Additional data collected.

Data	Time of Collection
Date and reason for visits (access and macroservices) provided by the health workers of the palliative care network	monitoring period
Hospital admissions and access to the emergency room	monitoring period
Indirect costs incurred by the family	end of the monitoring period (death or exit from the study)
Perception of the strengths and weaknesses of the care paths and palliative care assistance according to the personal experience of the caregivers (focus group)	end of the monitoring period (death or exit from the study)

**Table 2 healthcare-08-00221-t002:** General characteristics of the study population.

Variable	Category	*N*	(%)
Total, *N* (%)		1013	(100)
Sex, *N* (%)			
	Female	531	(52.4)
	Male	482	(47.6)
Age in years, mean (SD)		75.4	(13.0)
Healthcare setting, *N* (%)			
	Home Palliative Care Unit	524	(51.7)
	Hospital	260	(25.7)
	Hospice	229	(22.6)
Main diagnosis, *N* (%)			
	Cancer	865	(85.2)
	Lung	166	(16.4)
	Pancreas	76	(7.5)
	Colorectal	75	(7.4)
	Other clinical conditions	148	(14.6)
	Cardiovascular disease	36	(3.5)
	Dementia	26	(2.6)
	Respiratory diseases	25	(2.5)
Comorbidities, *N* (%)			
	None	416	(41.1)
	At least one	597	(58.9)

**Table 3 healthcare-08-00221-t003:** Main demanding problems observed in patients who accessed the local palliative care networks, stratified by diagnosis.

Variable	Cancer *N* = 865 (85.4%)	Noncancer *N* = 148 (14.6%)	Total *N* = 1013 (100%)	*p*-Value
Sex				
Male	418 (48.3)	64 (43.2)	482 (47.6)	0.30
Female	447 (51.7)	84 (56.8)	531 (52.4)	
Age, mean (SD)	74.3 (12.8)	82.0 (12.6)	75.4 (13.0)	<0.001
Severe emotional stress, *N* (%)	335 (38.7)	36 (24.3)	342 (63.4)	0.001
Extreme or severe clinical frailty, *N* (%)	115 (13.3)	65 (43.9)	180 (17.8)	<0.001
Completed ESAS questionnaire, *N*	780	64	844	
Anxiety, *N* (%)	456 (58.5)	33 (51.6)	489 (57.9)	0.35
Depression, *N* (%)	482 (61.8)	34 (53.1)	516 (61.1)	0.22
Dyspnea, *N* (%)	376 (48.2)	39 (60.9)	415 (49.2)	0.067
Pain, *N* (%)	523 (67.1)	33 (51.6)	556 (65.9)	0.017
Malaise, *N* (%)	614 (78.7)	45 (70.3)	659 (78.1)	0.16
Nausea, *N* (%)	348 (44.6)	22 (34.4)	370 (43.8)	0.15
Sleepiness, *N* (%)	543 (69.6)	46 (71.9)	589 (69.8)	0.81
Asthenia, *N* (%)	734 (94.1)	57 (89.1)	791 (93.7)	0.18

ESAS, Edmonton Symptom Assessment System.

**Table 4 healthcare-08-00221-t004:** Main demanding problems observed in patients who accessed the local palliative care networks, stratified by healthcare setting.

Variable	Home *N* = 524 (51.7%)	Hospice *N* = 229 (22.6%)	Hospital *N* = 260 (25.7%)	*p*-Value
Sex				
Male	258 (49.2)	104 (45.4)	120 (46.2)	0.54
Female	266 (50.8)	125 (54.6)	140 (53.8)	
Age, mean (SD)	75.5 (12.3)	76.1 (12.3)	74.5 (15.0)	0.38
Cancer, *N* (%)	473 (90.3)	191 (83.4)	201 (77.3)	<0.001
Severe emotional stress, *N* (%)	155 (29.6)	145 (63.3)	71 (27.3)	<0.001
Extreme or severe frailty, *N* (%)	68 (13.0)	69 (30.1)	43 (16.5)	<0.001
Completed ESAS questionnaire, *N*	479	168	197	
Anxiety, *N* (%)	274 (57.2)	105 (62.5)	110 (55.8)	0.39
Depression, *N* (%)	287 (59.9)	126 (75.0)	103 (52.3)	<0.001
Dyspnea, *N* (%)	240 (50.1)	82 (48.8)	93 (47.2)	0.79
Pain, *N* (%)	338 (70.6)	90 (53.6)	128 (65.0)	<0.001
Malaise, *N* (%)	389 (81.2)	116 (69.0)	154 (78.2)	0.004
Nausea, *N* (%)	239 (49.9)	57 (33.9)	74 (37.6)	<0.001
Sleepiness, *N* (%)	350 (73.1)	123 (73.2)	116 (58.9)	<0.001
Asthenia, *N* (%)	449 (93.7)	160 (95.2)	182 (92.4)	0.53

ESAS, Edmonton Symptom Assessment System.

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
