# Peer review of "Clinical Care Conditions and Needs of Palliative Care Patients from Five Italian Regions: Preliminary Data of the DEMETRA Project"

_healthcare, 2020, doi:10.3390/healthcare8030221_

Round 1
Reviewer 1 Report
This is a generally well written paper on the nature of palliative care in Italy. I wondered about whether the take-home messages of the paper could be made clearer and more interesting – for example, what are the consequences of your findings in terms of care delivery and resources needed? You present a number of tables of results but you make little of them. Specific point follow:
Page 2, line 47: should it be facing the problems (plural) associated with life-threatening illness?
Page 2, line 51: “currently around 40 million people would need palliative care” – how is this decided? Presumably by the WHO definition? It would be clearer to say that they qualified for palliative care using such a definition.
Page 2, line 57: “through hospital-territory integration” – I am not sure what this odd phrase means? Do you mean local amalgamation of hospital services?
Page 2, line 71: “this heterogenous situation” – how do you know the situation is heterogeneous ahead of time, before you have reported your results?
Page 3, line 92: I found it confusing that one of your exclusion criteria was those already in the palliative care network, since recruitment is done through network centres. This initially seemed like a contradiction, but I later realised that this means you are recruiting only new patients as they present for care to the palliative care network centres – hence your inclusion/exclusion criteria needs to make it clearer that you are only taking new patients into the study.
Page 3, line 100: “submitted to quality control” – what does this mean, and what does quality control entail?
Page 3, line 112: Can we please have a manufacturer and place of manufacture for all of your software tools used in this study. It would be clearer if you also briefly explained why you chose each tool and defined acronyms on first use. Some detail of how the tools work would also be useful in some cases – for example, what is the “surprise question” and how does it work?
Page 3, line 128: It would be better to put the bullet points for types of data you collected into a table.
Page 4, line 148: the sentence talking about “incident cases” is unclear. I can see it is part of a sample size estimate, but it would be clearer simply to talk about the number of patients entering palliative care each year, and then estimating how many you need to recruit in order to obtain a 4% margin or error. However, such estimates are based on a presumption of random selection (usually this is done for survey research) – hence patients entering your care centres are probably not be randomly selected? It would be useful therefore to make some comment in discussion on how typical the participating centres in your study are.
Page 4, line 157: It seems very odd to claim you are doing statistical testing on your qualitative data – and it is unclear which results stem from this.
Page 7, line 212: You make the point that palliative care is often thought to involve only those with cancer, and your results demonstrate this is not the case. You identify two cohorts of patients, presumably with different needs (e.g. their age and frailty differs), the smallest of which has cancer – yet you make little of what this means. What implications does this have on resource use, costs, required care expertise etc? Is this a recent change?
Page 7, line 234: You mention in discussion that there were missing responses and some dropouts in your study – this should be reported first in results – how many and % etc. In discussion you need to state this as a potential limitation.
END
Author Response
Answers to Reviewer #1
Comments and Suggestions for Authors
This is a generally well written paper on the nature of palliative care in Italy. I wondered about whether the take-home messages of the paper could be made clearer and more interesting – for example, what are the consequences of your findings in terms of care delivery and resources needed? You present a number of tables of results but you make little of them.
Dear Reviewer,
Thank you for revising our manuscript and for your appreciation that highlights the importance of our observational study evaluating the nature of palliative care in Italy.
Below you will find a point-by-point answer to each raised question. We hope this improved version of the manuscript could be considered suitable for publication on Healthcare.
Question: Page 2, line 47: should it be facing the problems (plural) associated with life-threatening illness?
Answer: Sentence has been correct as suggested.
Q: Page 2, line 51: “currently around 40 million people would need palliative care” – how is this decided? Presumably by the WHO definition? It would be clearer to say that they qualified for palliative care using such a definition.
A: The sentence has been written according to data from the WHO statement [reference 4] that stated that: “Each year, an estimated 40 million people are in need of palliative care, 78% of them people live in low- and middle-income countries…”. This statement is supported by data specifically obtained for palliative care needs [https://www.who.int/en/news-room/fact-sheets/detail/palliative-care] and, thus, we are strongly confident that they can be accurate for our purpose.
Q: Page 2, line 57: “through hospital-territory integration” – I am not sure what this odd phrase means? Do you mean local amalgamation of hospital services?
A: We apologize for this misunderstanding. With this sentence we wanted to affirm the importance of an integration between different levels of palliative care services as well as hospital healthcare, hospice services and home services in order to have greater efficiency in palliative care. The importance of a such organization is also stated by the World Health Organization [for further information please see at: https://apps.who.int/iris/bitstream/handle/10665/250584/9789241565417-eng.pdf;jsessionid=90AC192C917B4D3D81A3EDB5BC1CC931?sequence=1]
To better clarify the concept, in the manuscript we have updated the sentence in:
“…local palliative care networks to guarantee the continuity of care for the patient through the integration between hospital / hospice care and home care as also recommended by the WHO.”
Moreover, a new reference has been added according to this statement.
Q: Page 2, line 71: “this heterogenous situation” – how do you know the situation is heterogeneous ahead of time, before you have reported your results?
A: The sentence is supported by the Italian situation described upon in line 58-60, reference 7, where it is also illustrated how palliative care in Italy is not spread homogeneously in the nation.
However, as suggested, for greater clarity we have eliminated the term “heterogeneous” from the sentence and we have rewritten it accordingly.
Q: Page 3, line 92: I found it confusing that one of your exclusion criteria was those already in the palliative care network, since recruitment is done through network centres. This initially seemed like a contradiction, but I later realised that this means you are recruiting only new patients as they present for care to the palliative care network centres – hence your inclusion/exclusion criteria needs to make it clearer that you are only taking new patients into the study.
A: Thank you for this suggestion. We fully agree with you and apologize for this imprecise explanation. For further clarity, we therefore added the following inclusion criterion:
“1. new patients accessing the palliative care network during the recruitment period”
Q: Page 3, line 100: “submitted to quality control” – what does this mean, and what does quality control entail?
A: Thanks for the comment. With the term “quality control” we meant a check regarding the insertion of data by checking completeness and consistency of data (e.g. incorrect dates, data missing). We agree and we have integrated the sentence:
“submitted to quality control for evaluating completeness and consistency of recorded data”
Q: Page 3, line 112: Can we please have a manufacturer and place of manufacture for all of your software tools used in this study. It would be clearer if you also briefly explained why you chose each tool and defined acronyms on first use. Some detail of how the tools work would also be useful in some cases – for example, what is the “surprise question” and how does it work?
A: Thanks for your suggestion. We have modified the whole paragraph by adding more specific information on the tools and their origin, describing their characteristics in more detail.
Q: Page 3, line 128: It would be better to put the bullet points for types of data you collected into a table.
A: As required we have inserted the information in the bullet points into a new table (Table 1).
Q: Page 4, line 148: the sentence talking about “incident cases” is unclear. I can see it is part of a sample size estimate, but it would be clearer simply to talk about the number of patients entering palliative care each year, and then estimating how many you need to recruit in order to obtain a 4% margin or error. However, such estimates are based on a presumption of random selection (usually this is done for survey research) – hence patients entering your care centres are probably not be randomly selected? It would be useful therefore to make some comment in discussion on how typical the participating centres in your study are.
A: In order to clarify this concept, in the material and methods section we have changed the word “incident” with “new” (according to the inclusion criterion) and in the study limitations we have clarified that the use of a convenience sample could affect representativeness with respect to the Italian population with palliative care needs.
Q: Page 4, line 157: It seems very odd to claim you are doing statistical testing on your qualitative data – and it is unclear which results stem from this.
A: We apologize for this misunderstanding. For qualitative variables we did not intend data collected by qualitative methods but instead nominal, ordinal or dichotomous data.
Q: Page 7, line 212: You make the point that palliative care is often thought to involve only those with cancer, and your results demonstrate this is not the case. You identify two cohorts of patients, presumably with different needs (e.g. their age and frailty differs), the smallest of which has cancer – yet you make little of what this means. What implications does this have on resource use, costs, required care expertise etc? Is this a recent change?
A: Thank you for your comment. The biggest cohort we identified is the one of the patients affected by cancer. Nonetheless, these patients suffer from other conditions and comorbidities. Even if it is widely known that non-cancer patients and patients affected by not only cancer could benefit from palliative care approach [Gómez-Batiste et al. Palliat Med. 2017;31(8):754-763] in terms of quality of life, reduction of polypharmacy, reduction in hospitalization, fall prevention and reduction of discomfort, in Italy the main part of the cared ones are still affected by cancer. As we already showed in other study [Scaccabarozzi et al. J Palliat Med. 2018;21(5):631-637], early identification and service integration are substantial conditions to pursue the aim of expand palliative care.
Q: Page 7, line 234: You mention in discussion that there were missing responses and some dropouts in your study – this should be reported first in results – how many and % etc. In discussion you need to state this as a potential limitation.
A: We thank you for your comment and we apologize for our forgetfulness. We therefore added in the results, in describing Table 3 (past version Table 2), the following comment regarding the missing data in the ESAS (symptoms): “ESAS was completed in 844 cases (83.3% of the total sample). This was due to the inability of some patients to report on their symptoms given their serious condition.”
Reviewer 2 Report
Preliminary data from DEMETRA, a longitudinal, observational, non-interventional study, conducted in five Italian regions from May 2017 to November 2018 to determine clinical care conditions and needs of palliative care patients is presented. The following suggestions are offered in order to strengthen the manuscript.
Title: title is descriptive
Abstract: describes the major preliminary findings of the study.
Key words: consider the following changes to the list in order to improve MeSH term searchability: change "palliative care network" to "palliative care" and "network"; "patient's needs " to "assessment of healthcare needs"; "clinical frailty" to "adult, frail older"; and "emotional stress" to "stress, emotional." Alphabetize to: adult, frail older; assessment of healthcare needs; network; palliative care; stress, emotional
Lines 58-70: it is mentioned "at a national level [that] there are significant differences in the density of palliative care networks as well as the number of provided services and even the types of patients and their families who benefit from these services." A paragraph about the differences between end of life vs. palliative care in the Italian context would be helpful here because palliative care is a broader term, and it seems that the discussion is more about end-of-life care, which is different.
Lines 101-137: it is very likely that all of these patients received pharmacotherapy. An assessment of their drug therapies in terms of drug-related needs and drug therapy problems would provide a more complete picture of their overall health needs and care provided. Pharmacotherapy for many patients in palliative care can overwhelm them, and there are a significant number of patients with anxiety, depression, and pain in your sample which may indicate sub-optimal drug therapy. See Abernethy AP. Palliative care pharmacotherapy literature summaries and analyses. J Pain Palliat Care Pharmacother. 2009;23(2):174-181. doi:10.1080/15360280902901404 for additional information on this aspect.
Lines 111-122: there is no frailty scale used to determine levels of frailty. The ESAS doesn't measure frailty. How was frailty measured? This is a major limitation that must be corrected.
Line 135: you mention gathering data from a focus group, but do not present any data, or describe how the data were analyzed. Please omit or expand.
line 160 - omit.
references: not in mdpi style.
Author Response
Answers to Reviewer #2
Preliminary data from DEMETRA, a longitudinal, observational, non-interventional study, conducted in five Italian regions from May 2017 to November 2018 to determine clinical care conditions and needs of palliative care patients is presented. The following suggestions are offered in order to strengthen the manuscript.
Dear Reviewer, thank you for revising our manuscript and for your suggestions and comments.
Question: Title: title is descriptive. Abstract: describes the major preliminary findings of the study.
Key words: consider the following changes to the list in order to improve MeSH term searchability: change "palliative care network" to "palliative care" and "network"; "patient's needs " to "assessment of healthcare needs"; "clinical frailty" to "adult, frail older"; and "emotional stress" to "stress, emotional." Alphabetize to: adult, frail older; assessment of healthcare needs; network; palliative care; stress, emotional.
Answer: Thank you for this suggestion. Keywords have been updated according to your recommendations.
Q: Lines 58-70: it is mentioned "at a national level [that] there are significant differences in the density of palliative care networks as well as the number of provided services and even the types of patients and their families who benefit from these services." A paragraph about the differences between end of life vs. palliative care in the Italian context would be helpful here because palliative care is a broader term, and it seems that the discussion is more about end-of-life care, which is different.
A: We appreciate your suggestion. As reported in the institutional report that we mentioned in our paper, there are few evidences of the differences between EOL care vs palliative care. But, as we have showed in an another study [Scaccabarozzi et al. J Palliat Med. 2018;21(5):631-637] early identification and services integration are two of the main conditions needed to provide palliative care paths and avoid the flattening on EOL care in those situations where palliative care would be preferred. As we recorded in another study [Scaccabarozzi et al., Healthcare 2019], whose aim was to monitoring the activities of the Italian Home Palliative Care Services, the extension of the palliative care services provided to frail non-cancer and paediatric patients, affected by complex and advanced chronic conditions, is still inadequate in Italy.
Q: Lines 101-137: it is very likely that all of these patients received pharmacotherapy. An assessment of their drug therapies in terms of drug-related needs and drug therapy problems would provide a more complete picture of their overall health needs and care provided. Pharmacotherapy for many patients in palliative care can overwhelm them, and there are a significant number of patients with anxiety, depression, and pain in your sample which may indicate sub-optimal drug therapy. See Abernethy AP. Palliative care pharmacotherapy literature summaries and analyses. J Pain Palliat Care Pharmacother. 2009;23(2):174-181. doi:10.1080/15360280902901404 for additional information on this aspect.
A: Unfortunately, data about pharmacotherapy have been not collected in this study since it was not a main aim of our investigation. However, we have added this limitation in the Discussion section and added the reference for clarifying why this information could be of interest for our analysis.
Q: Lines 111-122: there is no frailty scale used to determine levels of frailty. The ESAS doesn't measure frailty. How was frailty measured? This is a major limitation that must be corrected.
A: Thanks for the comment, your annotation is correct, the fragility data is not obtained from ESAS but, as described in row 119, to report the presence of a complex fragility we used the specific item present in the NECPAL sheet, which exactly is:
markers of severity and extreme frailty (Yes if t least 2 of the following in the last 6 months):
- Persistent pressure ulcers (stage III-IV)
- Recurrent infections (>1)
- Delirium
- Persistant Disphagia
- Falls (>2)
For clarity we have also added the sentence:
“The specific item of frailty of NECPAL has been used to define the patient in a condition of extreme fraility”
Q: Line 135: you mention gathering data from a focus group, but do not present any data, or describe how the data were analyzed. Please omit or expand.
A: We would thank you for your suggestion. We are still analysing data from the focus groups and, thus, these analyses are currently not available. However, according to the high quantity of collected data, we would prefer to analyse and discuss our findings in another paper to dedicate to focus groups evidences. However, if requested from you or required by the Editor we will reconsider this opportunity.
Q: line 160 - omit.
A: Line removed
Q: references: not in mdpi style.
A: Thank for this suggestion, we updated the references according to the mdpi style
Round 2
Reviewer 2 Report
The authors gave adjusted the paper by providing the reader with additional details about their interesting methodology and findings. I agree that the focus group findings are best presented in another manuscript. Thank you for the opportunity to comment on your fine work.